# Knockout of the neonatal Fc receptor alters immune complex trafficking and lysosomal function in cultured podocytes

**George Haddad[1], James Dylewski[1,2], River Evans[1], Linda Lewis[1], Judith Blaine[1]\***

**1** Renal Division, Department of Medicine, University of Colorado Anschutz Medical Campus, Aurora, CO, United States of America, **2** Department of Nephrology, Denver Health Medical Center, Denver, CO, United States of America

* Judith.Blaine@cuanschutz.edu

**Data Availability Statement:** All relevant data are within the paper and its Supporting Information files.

## Abstract

Podocytes are key to preventing the filtration of serum proteins into the urine. Recent evidence also suggests that in immune mediated kidney diseases, podocytes are the targets of immune complexes (ICs). The mechanisms whereby podocytes handle and respond to ICs remain unknown. The neonatal Fc receptor (FcRn) is involved in IgG handling in podocytes and is also required in dendritic cells to traffic ICs to the lysosome for proteolytic degradation of antigen and presentation on MHC II. Here we examine the role of FcRn in handling ICs in podocytes. We show that knockout of FcRn in podocytes results in decreased trafficking of ICs to the lysosome and increases IC trafficking to recycling endosomes. FcRn KO also alters lysosomal distribution, decreases lysosomal surface area and decreases cathepsin B expression and activity. We demonstrate that signaling pathways in cultured podocytes differ after treatment with IgG alone versus ICs and that podocyte proliferation in both WT and KO podocytes is suppressed by IC treatment. Our findings suggest that podocytes respond differentially to IgG versus ICs and that FcRn modifies the lysosomal response to ICs. Elucidating the mechanisms underlying podocyte handling of ICs may provide novel pathways to modulate immune mediated kidney disease progression.

## Introduction

Proteinuria is a hallmark of many renal diseases including immune mediated kidney diseases and an increase in proteinuria predicts renal disease progression [1]. Under normal conditions, the glomerular filtration barrier which consists of glomerular endothelial cells, the glomerular basement membrane and podocytes prevents leakage of serum proteins the size of albumin or larger into the urine. Podocytes which are terminally differentiated cells with a complex cytoarchitecture play a key role in preventing proteinuria [2]. Podocyte injury leads to foot process effacement, cell detachment and death, resulting in worsening glomerular disease and loss of kidney function [3, 4]. Podocytes are often the targets of immune mediated kidney disease and recent studies of glomerulonephridites including lupus nephritis, minimal change disease and membranous nephropathy have shown that podocytes are the targets of

**Funding:** The work in this manuscript was was funded by NIH National Institute of Diabetes and Digestive and Kidney Diseases R01 DK104264 to JB. The funders had no role in study design, data collection and analysis, decision to publish, or preparation of the manuscript.

**Competing interests:** The authors have declared that no competing interests exist.

autoantibodies and that the podocyte response to immune complexes mediates disease outcome [5–7]. The mechanisms underlying how podocytes handle immune complexes, however, remain unknown.

The neonatal Fc receptor (FcRn), a major histocompatibility complex (MHC) class I related receptor, is expressed in various cells including endothelial cells, podocytes and dendritic cells and is necessary for the transcytosis of monomeric IgG and albumin leading to the extended half-lives of these proteins [8]. In addition, FcRn is required in dendritic cells, monocytes and macrophages to direct immune complexes (ICs) to the lysosome for proteolytic degradation of antigen and presentation on MHC II [9]. While it is known that FcRn mediates lysosomal trafficking of immune complexes in dendritic cells, its role in trafficking of ICs in podocytes has not been investigated. Here, using wild type (WT) and FcRn knockout (KO) podocytes we investigate the role of FcRn in lysosomal trafficking of ICs. Lysosomes have traditionally been viewed as the "garbage disposal" units of the cell. Increasing evidence, however, demonstrates that lysosomes are dynamic and metabolically active organelles that play an important role in cellular signaling and homeostasis [10–12]. Here we show that knockout of FcRn in cultured podocytes alters lysosomal size and function and cathepsin B expression and activity. In addition we find that in cultured podocytes signaling pathways and cellular proliferation are differentially upregulated in response to either monomeric IgG or immune complexes. Our work demonstrates that FcRn mediated trafficking of immune complexes modulates lysosomal function in podocytes. and that cultured podocytes respond differently to IgG versus ICs.

## Materials and methods

### Podocyte cell culture

Podocytes were isolated from WT or global FcRn KO mice as previously described [13] and then conditionally immortalized using a thermosensitive SV40 antigen. Briefly, podocytes were cultured on type I collagen coated flasks or dishes. After transformation, replication occurred at 33˚C. To induce differentiation, the podocytes were cultured at 37˚C for 8–10 days. All experiments were performed using differentiated and confluent podocytes. To examine cathepsin B expression or lysosomal function, WT and FcRn KO cells were treated with regular media (RPMI 1640 + 10% fetal bovine serum) or regular media plus immune complexes (ICs). Immune complexes were made by incubating 4-Hydroxy-3-nitrophenylacetyl (NP)-ovalbumin (NP-ova, concentration 20 μg/ml, Biosearch technologies, cat. # N-5051) with anti NP-ova IgG1 antibody (20 μg/ml). The anti NP-ova IgG1 antibody (clone N1G9) was a kind gift of Dr. Raul Torres (University of Colorado).

### Immunofluorescence

Differentiated WT or FcRn KO podocytes were fixed in 4% paraformaldehyde. After fixation, podocytes were rinsed with PBS, permeabilized in 0.1% Triton-X100 in PBS (PBS-X), blocked in 5% BSA in PBS-X and incubated overnight at 4˚C with the following primary antibodies depending on the experiment: rat anti-mouse LAMP1 (1:500, Santa Cruz, cat. # sc-19992), goat anti-mouse IgG (1: 100, Vector Labs, cat. #NC0207995), rat anti-mouse IgG1 (1:100, Biolegend, cat. # 50169164), rabbit anti-mouse Rab11 (1:200, Cell Signaling, cat. # 5589), rabbit anti-mouse Rab7 (1:200, Cell Signaling, cat. #9367). Podocytes were rinsed and incubated with the appropriate secondary antibodies (Alexa 568 goat anti-rat, Alexa 488 goat anti-rat, Alexa 488 goat anti-rabbit, Alexa 568 donkey anti-goat, Alexa 568 goat anti-rabbit, Invitrogen). Hoechst was used to stain nuclei and Alexa 635 phalloidin was used to stain actin. Images were acquired using a Zeiss LSM 780 microscope. Colocalization between immune complexes and

LAMP1 or ICs and Rab11 was evaluated using the Coloc2 Tool in ImageJ (NIH). Lysosomal area was also assessed using ImageJ.

## Western blot

WT or FcRn KO podocytes were treated with regular media, anti NP-ova IgG1 antibody (clone N1G9, 20 μg/ml), NP-ova (20 μg/ml) or ICs for 4 or 48 hours. Cell lysates were harvested in RIPA buffer (Thermo Fisher, cat. # 89900) supplemented with phosphatases and proteases inhibitors, subject to western blot analysis and probed with rat anti mouse LAMP1 (Santa Cruz, cat. #sc-19992) or rabbit anti mouse cathepsin B (Cell Signaling, cat. #31718) antibodies. ß-actin (Sigma, cat. # A1978) was used as a loading control. Image J was used to quantitate band intensity.

For signaling studies, WT or FcRn KO podocytes were not treated or treated with NP-ovalbumin, IgG or immune complexes as above for the indicated times. Cell lysates were harvested in RIPA buffer, subject to western blot analysis and probed with rabbit ant mouse AKT (cell Signaling, cat # 4691), rabbit anti mouse phospho-AKT (S473, Cell Signaling, cat. # 4060), rabbit anti mouse ERK1/2 (Cell Signaling, cat. # 4695) or rabbit anti mouse phospho-ERK1/2 (Cell Signaling, cat. # 4377). ß- actin was used as a loading control and band intensity was quantitated using ImageJ.

## Lysotracker staining

WT or FcRn KO podocytes were treated with regular media or immune complexes for 4 hours and then stained with 50 nM Lysotracker Red DND99 (Invitrogen, cat. #L7528) at 37˚C for 15 minutes and then rinsed and imaged immediately using a Zeiss 780 LSM confocal microscope. Corrected total cell fluorescence (CTCF) was measured using Image J.

## Fluorescence recovery after photobleaching (FRAP) experiments

FRAP was performed to determine active cathepsin B activity in live podocytes as described by Pryor [14]. Briefly, podocytes were grown on collagen coated dishes with coverslip bottoms (MatTek). Cells were treated with regular media or regular media + immune complexes as described above. After 4 or 48 hours of treatment, podocytes were rinsed and incubated for 30 minutes in a 1X solution of Magic Red Cathepsin B (Abcam, cat. #270772) according to the manufacturer's instructions. During visualization of podocytes, a region of interest (ROI) containing multiple lysosomes was defined. Three images of the ROI were taken prior to bleaching. The ROI was bleached with 100 iterations using the 568 laser for 5 seconds to reduce ROI baseline fluorescence intensity by at least 80%. The ROI was then continuously scanned until the fluorescence in the ROI recovered to a plateau. Fluorescence intensity relative to peak intensity (y-axis) was plotted versus time (x-axis). A nonlinear regression curve with one-phase decay was fit to the data enabling determination of half-time ($t_{1/2}$) of recovery using GraphPad Prism 9 software.

## Cell proliferation

Cell proliferation was determined using Click-iT EdU proliferation assay (Invitrogen, cat. # C10499) as per manufacturer's instruction. Briefly, 15000 cells seeded into a 96-well tissue culture plate and treated with normal growth medium (10% FBS), PBS, anti NP-ova IgG1 antibody (clone N1G9, 20 μg/ml), NP-ova (20 μg/ml) or ICs for 4h. Ten μM EdU was added to each well and the cells were incubated overnight at 37˚C, 5% CO2. The next day the EdU

incorporation was determined as described in the manual and the plates were developed and read using a fluorescent plate reader (excitation 568 nm and emission 585 nm).

### Statistical analyses

All data were analyzed by using GraphPad Prism version 9.3.0 (GraphPad Software, San Diego, CA). The results are presented as the mean ± SEM. Statistical analysis was performed using t-tests for two groups and one-way analysis of variance for 3 or more groups. Tukey's post hoc t-test was applied to the ANOVA data. Statistical significance was defined as $p < 0.05$.

## Results

### Trafficking of ICs to the lysosome is decreased and trafficking of ICs to recycling endosomes is increased in FcRn KO podocytes

In dendritic cells, FcRn is required to traffic immune complexes to the lysosome for processing [9]. To examine whether FcRn is required to direct ICs to the lysosome in podocytes, we treated WT and FcRn KO podocytes with immune complexes for 4 hours and examined colocalization between ICs and the lysosomal compartment using confocal microscopy. We found that in FcRn KO podocytes, there was significantly less colocalization between ICs and LAMP1-positive (lysosomal) compartments (Fig 1A), suggesting decreased trafficking of ICs to the lysosomes in the FcRn KO podocytes. We hypothesized that in FcRn KO podocytes, ICs might preferentially be diverted to recycling endosomes. We used Rab11 as a recycling endosome marker and found increased colocalization of ICs and Rab11-positive compartments in FcRn KO podocytes compared to wild type (Fig 1B), suggesting preferential trafficking of ICs to recycling endosomes in the KO cells.

### Lysosomal size and cellular position differ in WT and FcRn KO podocytes

Given the differences in IC trafficking to the lysosome in WT and FcRn KO podocytes, we examined lysosomal size and position in these cells. Previous studies have shown that lysosomal function is modulated by the size and position of lysosomes within the cell, with larger more mature and proteolytically active lysosomes being found closer to the cell nucleus [12, 15, 16]. We examined lysosomal area in WT and FcRn KO podocytes at baseline and after treatment with ICs. We found that at baseline, lysosomal area was significantly decreased in FcRn KO podocytes compared to WT (Fig 2A), suggesting that in the FcRn KO cells, lysosomes are less active. After treatment with ICs for 4 hours, WT podocytes had a non-significant increase in surface area compared to baseline conditions. In contrast, after IC treatment FcRn KO podocytes had a significant increase in lysosomal area compared to baseline although the average lysosomal area was still significantly decreased compared to WT podocytes treated with ICs (Fig 2A).

Lysosomal activation results in migration of lysosomes from more peripheral cellular locations to the perinuclear region [17–19]. Given that lysosomes in FcRn KO podocytes have decreased surface area compared to WT suggesting altered lysosomal function in FcRn KO podocytes, we hypothesized that their cellular location in response to an immune challenge would be different. To test this hypothesis, we examined lysosomal distribution in WT and FcRn KO podocytes after an immune challenge. After treatment with ICs for 4 hours, lysosomes in WT podocytes clustered around the nucleus (arrows) whereas in FcRn KO cells, lysosomes remained distributed throughout the cell (Fig 2B), suggesting that treatment with ICs did not result in lysosomal activation in the knockout.

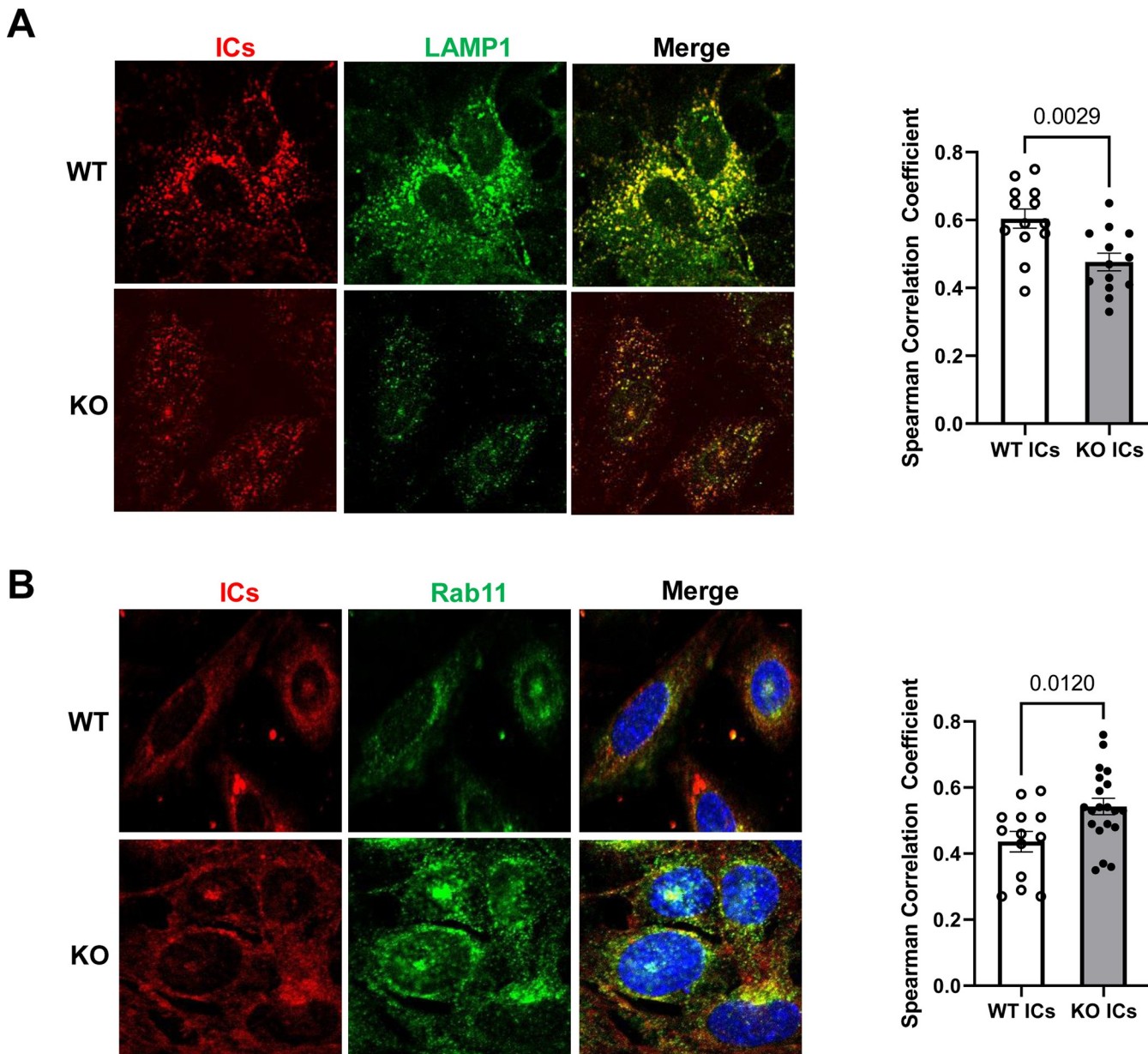

**Fig 1. Knockout of FcRn in cultured podocytes significantly decreases immune complex trafficking to the lysosome and increases IC trafficking to recycling endosomes.** A, Colocalization between lysosomes (LAMP1+ compartments) and immune complexes in WT or KO podocytes. B, Colocalization between recycling endosomes (Rab11+ compartments) and immune complexes in WT or KO podocytes. Each point on the graph represents the average correlation coefficient for 2–4 cells from n = 4 independent experiments.

## Colocalization of lysosomes with late endosomal compartments is altered in FcRn KO podocytes

Since trafficking of exogenous proteins to the lysosome involves transit through the late endosome [20], we sought to determine if colocalization of lysosomes and late endosomes was altered in KO podocytes. We examined lysosomal colocalization with late endosomal (Rab7 positive) compartments. After treating podocytes with ICs for 4 hours, we found that there was significantly less colocalization between late endosomal (Rab7+) compartments and lysosomal (LAMP1+) compartments in the KO (Fig 3A).

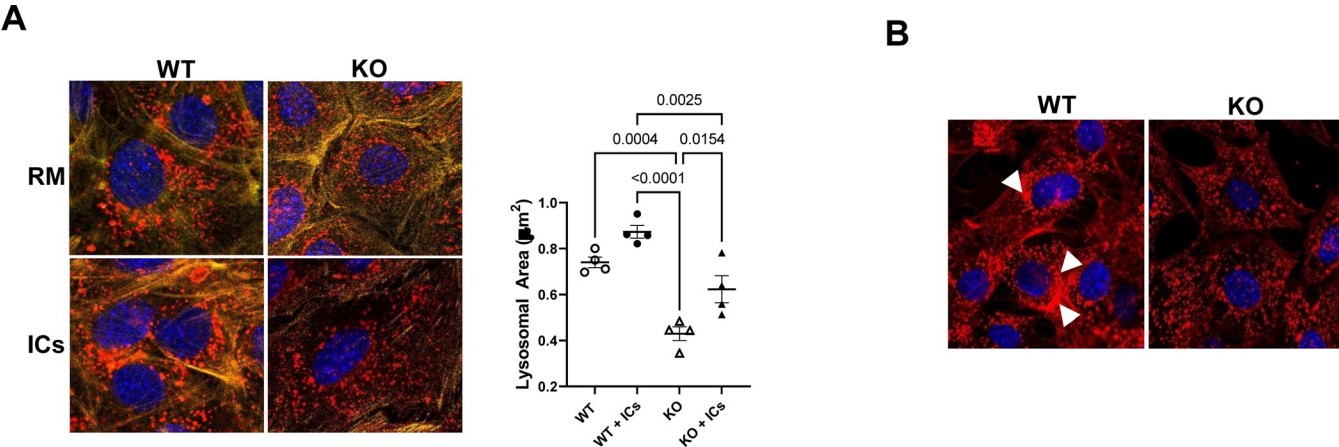

**Fig 2. Lysosomal area is significantly decreased and lysosomal distribution is altered in FcRn KO podocytes compared to WT after treatment with immune complexes.** *A*, Lysosomal area is decreased in KO podocytes at baseline and after treatment with ICs. RM, regular media; ICs, immune complexes. Lysosomes are stained red, nuclei are blue, actin is yellow. Symbols represent mean surface area for lysosomes in 20 different cells per experiment, n = 4 independent experiments. *B*, Treatment with ICs results in perinuclear clustering of lysosomes in WT podocytes (arrows) but not KO. Lysosomes are stained red.

## FcRn KO results in fewer acidic compartments in podocytes and alterations in LAMP1 and cathepsin B activity in podocytes

We next examined lysosomal acidity and expression of the lysosomal membrane protein LAMP1 as well as one of the key lysosomal enzymes, cathepsin B. Lysotracker Red is a cell permeant dye that preferentially accumulates in acidic compartments in living cells [21]. To examine acidic compartments in podocytes in real time, we stained WT and FcRn KO podocytes at baseline and after treatment with ICs with Lysotracker Red. We found that at baseline Lysotracker Red staining was significantly decreased in KO podocytes compared to WT (Fig 4A). Treatment with ICs increased Lysotracker red staining in KO podocytes but staining was

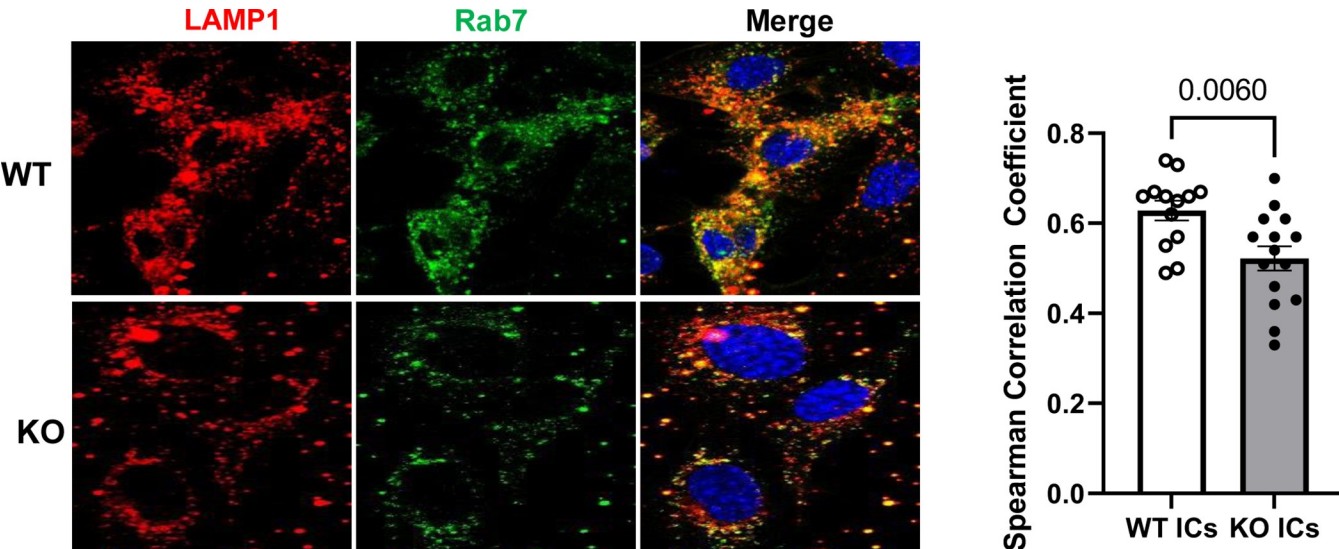

**Fig 3. FcRn KO significantly reduces colocalization between late endosomal and lysosomal compartments after treatment with ICs.** Late endosomes are Rab7+ and are stained red. Lysosomes are LAMP1 positive and are stained green. Each point on the graph represents the average correlation coefficient for 3–5 cells from n = 3 independent experiments.

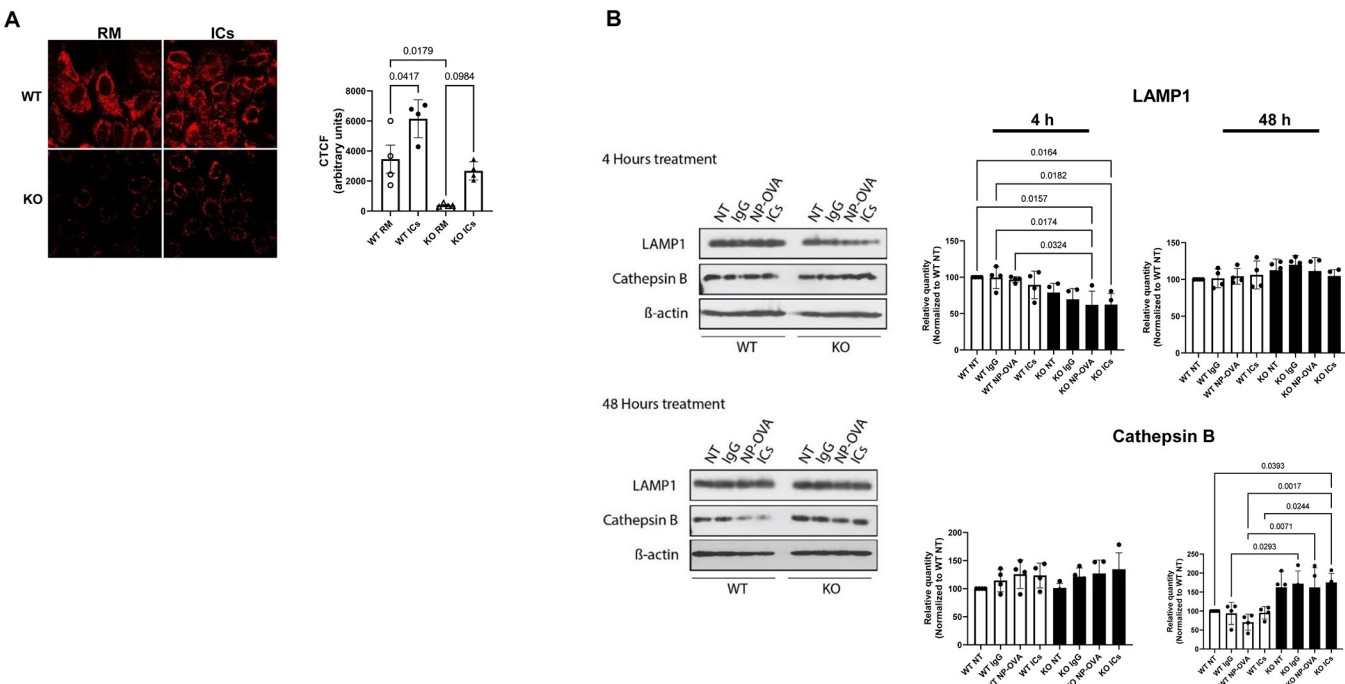

**Fig 4. FcRn KO alters lysotracker staining and LAMP1 and cathepsin B expression in podocytes.** *A*, Lysotracker Red staining is significantly decreased in WT and FcRn KO podocytes at baseline and after treatment with ICs. RM, regular media; ICs, immune complexes; CTCF, corrected total cell fluorescence. Symbols represent mean CTCF for 20 different cells, n = 4 independent experiments. *B*, Western blot analysis of LAMP1 and cathepsin B expression in WT and FcRn KO podocytes treated with IgG alone, NP-ovalbumin or immune complexes for 4 or 48 hours, n = 3 independent experiments.

still significantly decreased compared to that in WT podocytes treated with ICs, suggesting decreased lysosomal activation in KO cells.

Given the differences in lysosomal appearance and cellular location, we examined the expression of LAMP1, a lysosomal marker, and cathepsin B, a cysteine protease specific to lysosomes, in WT and FcRn KO podocytes. We treated podocytes with ICs for 4 hours and 48 hours to mimic acute and chronic exposure to an immune challenge. We found that in WT podocytes, LAMP 1 expression was unchanged across any treatment condition at 4 hours (Fig 4B). Treatment with NP-ovalbumin for 4 hours, resulted in a significant decrease in LAMP1 in KO podocytes compared to WT (Fig 4B). Treatment with ICs for 4 hours also led to a significant decrease in LAMP1 expression in the KO compared to untreated WT cells. After 48 hours of treatment, there was no difference in LAMP1 expression for WT versus KO for any of the treatment conditions (Fig 4B), suggesting that KO podocytes upregulated expression of LAMP1 protein after sustained treatment with NP-ovalbumin or immune complexes.

In contrast, treatment of WT and KO podocytes with IgG, NP-ovalbumin or immune complexes did not result in any significant change in cathepsin B expression after 4 hours (Fig 3D). However, by 48 hours, treatment with IgG, NP-ovalbumin or immune complexes resulted in a significant decrease in cathepsin B expression in WT podocytes compared to KO (Fig 3D), suggesting that sustained treatment led to downregulation of cathepsin B expression at the protein level in WT podocytes.

## Cathepsin B activity is significantly decreased in FcRn KO podocytes after an immune challenge

Based on the differences in lysosomal size, location and cathepsin B expression at the protein level, we hypothesized that there would also be a difference in cathepsin B activity in WT

versus FcRn KO podocytes. We used Cathepsin B Magic Red as this reagent allows determination of the activity of this enzyme in lysosomes in living cells in real time [14]. To examine cathepsin B activity after treatment with immune complexes, WT or FcRn KO podocytes were treated with ICs for 4 or 48 hours. Cathepsin B activity was measured in live cells using fluorescence recovery after photobleaching (FRAP) as previously described [14, 22]. Cathepsin B Magic Red is a fluorophore coupled to a substrate specific for cathepsin B. When the substrate is cleaved by cathepsin B, the fluorophore is released and fluoresces after being excited by the appropriate wavelength. FRAP was performed by bleaching a region of interest containing a lysosome and then the time to recovery of the Magic Red signal was measured (Fig 5A). The intensity of the Magic Red signal over time relative to the initial intensity was graphed. Representative curves for each condition are shown in Fig 4B. The half-time ($t_{1/2}$) of fluorescence recovery is indirectly proportional to cathepsin B activity–i.e. shorter $t_{1/2}$ to recovery indicates increased cathepsin B activity. After treatment with ICs, FcRn KO podocytes had significantly longer $t_{1/2}$ to recovery compared to WT podocytes at both 4 hours and 48 hours, indicating decreased cathepsin B activity in the FcRn KO cells (Fig 5C). Interestingly, while $t_{1/2}$ recovery was significantly increased in the KO podocytes at both 4 and 48 hours, overall $t_{1/2}$ recovery was increased in both WT and KO podocytes at 48 hours compared to 4 hours, suggesting an overall decrease in cathepsin B activity with prolonged exposure to ICs.

## Signaling pathways differ between WT and FcRn KO podocytes after treatment with ICs

To examine cell survival signaling pathways after treatment with immune complexes, we studied Akt and phospho-Akt expression at baseline and after treatment with IgG, NP-ova or immune complexes. Phosphorylation of Akt at serine 473 (S473) upregulates cell metabolism and growth and is involved in antigen presentation in an immune response [23, 24]. There

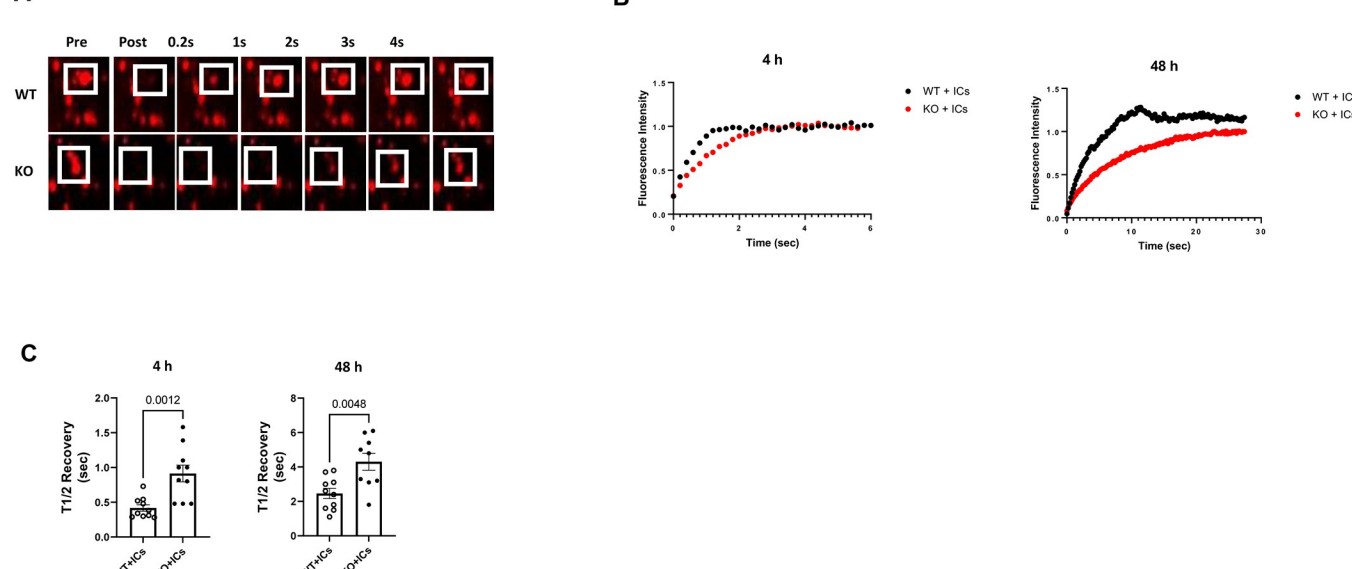

Fig 5. Cathepsin B activity is significantly reduced in FcRn KO podocytes after treatment with ICs. *A*, Representative pre and post bleach images for WT or KO podocytes treated with ICs. Square indicates the bleached area. *B*, Representative post bleach fluorescence intensity versus time curves for WT or FcRn KO podocyte treated with ICs for 4 hours or 48 hours. *C*, $T_{1/2}$ recovery for WT or KO podocytes treated with ICs for 4 hours or 48 hours, n = 4 independent experiments.

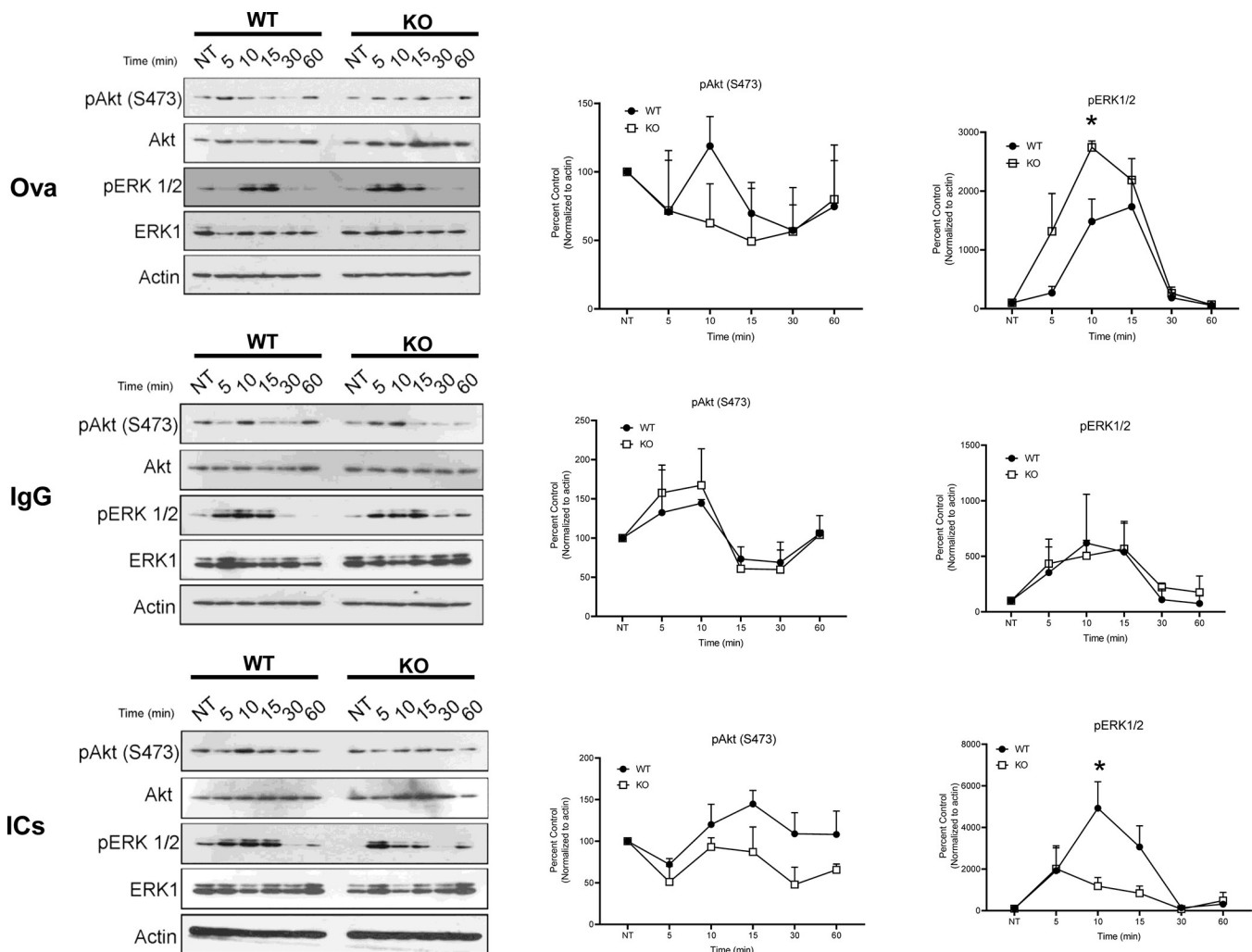

**Fig 6. Western blot analysis of Akt, phospho-Akt, ERK1/2 and phospho-ERK1/2 expression in WT and FcRn KO poodcytes treated with NP-ovalbumin (Ova), IgG or immune complexes (ICs).** *, P < 0.05, n = 3 independent experiments.

was no increase in phospho-Akt expression compared to baseline after treatment with ICs in KO podocytes whereas WT podocytes showed increased pAkt expression compared to baseline starting 10 minutes after treatment (Fig 6). Similarly, KO podocytes treated with NP-ovalbumin did not show an increase in p-Akt expression compared to baseline whereas WT podocytes had increased p-Akt expression compared to baseline at 10 minutes (Fig 6). In contrast, both WT and KO podocytes had increased pAkt expression compared to baseline after treatment with IgG.

We also examined the extracellular signal-regulated kinase 1 and 2 (ERK1/2) pathway after treatment of WT or FcRn KO podocytes with IgG, NP-ovalbumin or ICs. Phosphorylation of ERK1/2 is a critical step in promoting G1 to S transition and upregulating cell proliferation [25]. ERK1/2 signaling is also involved in the immune response [26]. We found that phospho-ERK1/2 was significantly upregulated in WT podocytes compared to KO podocytes 10 minutes after treatment with ICs. In contrast to IC treatment, however, we found that after treatment with NP-ovalbumin, phospho-ERK1/2 was significantly upregulated in KO podocytes at 10 minutes compared to WT (Fig 6). The different signaling responses to IgG alone versus

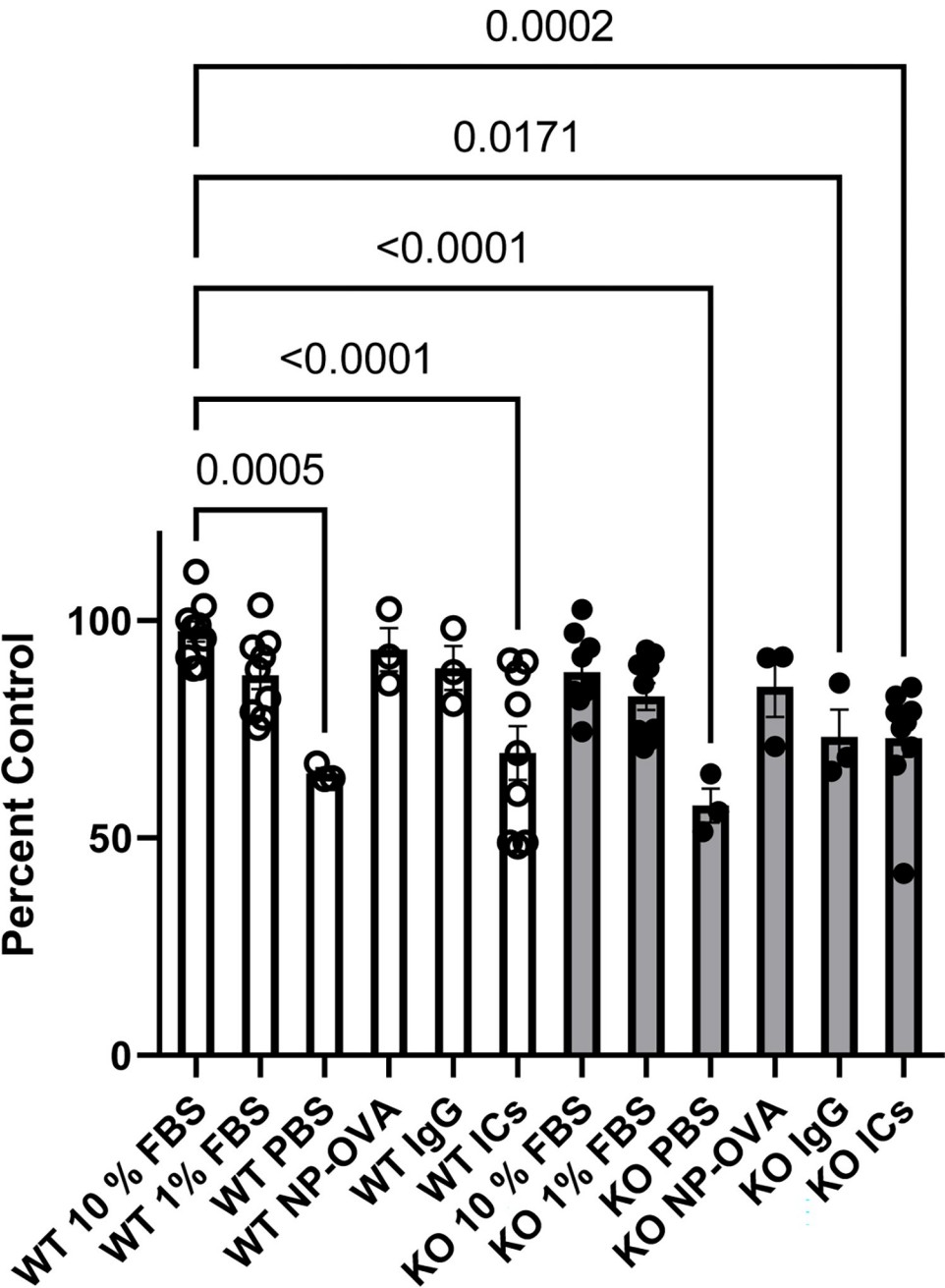

**Fig 7. Treatment with ICs causes a significant decrease in podocyte proliferation.** EdU proliferation assay values expressed as a percent of control (WT 10% FBS).

immune complexes suggests that podocytes have the ability to distinguish monomeric IgG from ICs.

To further investigate cell proliferation we performed an EdU assay. As expected, we found that growing cells in phosphate buffered saline (PBS) caused a significant reduction in cell proliferation (Fig 7). We found that at baseline, KO cells grown in regular media with 10% FBS (standard conditions) had a reduction in cell proliferation compared to WT cells grown in 10% FBS. Interestingly, treatment with immune complexes caused a significant reduction in

proliferation in both WT and KO podocytes compared to WT podocytes grown in standard conditions. In KO podocytes, treatment with IgG alone also caused a significant reduction in proliferation compared to WT podocytes grown in standard conditions.

## Discussion

Podocytes are a key part of the glomerular filtration barrier where they serve to prevent leakage of serum proteins such as albumin and IgG into the filtrate. Podocytes have also been proposed to act as immune cells [27] and express certain receptors associated with the immune system such as the neonatal Fc receptor which is required for antigen presentation in dendritic cells [9], CD80, a co-receptor necessary for T cell activation and Toll-like receptor 4 (TLR4) which is involved in the innate immune response [28]. In addition, knockout of MHC II in podocytes has been shown to attenuate immune-mediated nephritis [29] and knockout of Twist1, a key transcription factor which limits recruitment of inflammatory cells, selectively in podocytes has been shown to significantly worsen immune-mediated nephritis [30]. The precise mechanisms whereby podocytes participate in immune functions remain to be determined.

Lysosomes are key players in the immune system as trafficking of antigen/antibody complexes to the lysosome is required for antigen presentation and altered levels of lysosomal enzymes have been found in several autoimmune diseases including rheumatoid arthritis and systemic lupus erythematosus [17]. To date, little is known about lysosomal function in podocytes after an immune challenge. Here we examined lysosomal function in cultured wild type and FcRn KO podocytes at baseline and after an immune challenge. In accordance with the requirement for FcRn to traffic immune complexes to the lysosome, we found that there was significantly less colocalization of immune complexes and LAMP1, a lysosomal marker, in FcRn KO podocytes compared to WT. In addition, we found that at baseline, lysosomal area was significantly decreased in FcRn KO podocytes and that this decrease in lysosomal area persisted after an immune challenge. We also found that after an immune challenge there was a significant difference in lysosomal localization, with lysosomes clustering around the nucleus in WT podocytes and localizing to the periphery in FcRn KO podocytes. Since previous studies have shown that increased lysosomal size and perinuclear clustering is associated with lysosomal activation, our data suggest that FcRn knockout in podocytes decreases lysosomal activity. In other cell types, perinuclear localization of lysosomes increases cathepsin activity, one of the major types of hydrolytic enzymes within lysosomes [18, 19]. To examine cathepsin B activity in WT and FcRn KO podocytes directly after an immune challenge, we used FRAP and found that FcRn KO podocytes had a significant reduction in cathepsin B activity after treatment with immune complexes both at 4 hours and 48 hours after treatment. Examination of LAMP1 expression by Western blot analysis showed that FcRn KO podocytes had decreased LAMP1 expression 4 hours after IC treatment compared to WT podocytes but that LAMP1 expression at 48 hours was equivalent to WT. The mechanism underlying this finding is unclear but may be due to KO podocytes trying to compensate over time for baseline alterations in lysosomal function. In contrast, cathepsin B expression by Western blot was similar in WT and KO podocytes 4 hours after treatment with ICs but was significantly decreased in WT podocytes at 48 hours compared to KO. One hypothesis to explain these findings is that in WT podocytes, a sustained increase in cathepsin B activity leads to a reduction in enzyme protein levels over time. Taken together, these findings suggest that knockout of FcRn has a significant effect on lysosomal function in podocytes.

Recent findings have overturned the long held view that lysosomes are simply the "garbage disposal" units of the cell. Lysosomes are now known to be dynamic organelles whose size and position in the cell vary depending on cellular needs and stresses [10]. Lysosomes also

communicate with other organelles and have been found to play an important role in several cellular processes including metabolic signaling, immunity, cell adhesion and apoptosis [31]. While knockout or genetic defects in lysosomal enzymes have been shown to alter lysosomal activity [32–34], to our knowledge this is the first report of knockout of a trafficking protein causing changes in lysosomal function. Our findings show that decreasing delivery of immune complexes to lysosomes alters lysosomal size and cathepsin activity which suggests that there is a feedback mechanism between cargo trafficking to the lysosome in podocytes and lysosomal structure and function. Additional experiments are needed to dissect the mechanisms underlying these findings but modulation of lysosomal function may provide therapeutic targets to ameliorate disease severity in immune mediated kidney diseases. Indeed, Plaquenil, an agent that inhibits lysosomal function by increasing lysosomal pH [35] has long been used as a mainstay of treatment for immune-mediated diseases such as lupus and rheumatoid arthritis. An understanding of how immune complexes are trafficked to the lysosome in podocytes and how an immune insult may affect lysosomal function is of increased relevance given recent studies demonstrating that IgG or immune complex uptake by podocytes modulates immune mediated kidney diseases including lupus nephritis and membranous nephropathy [5, 6]. Furthermore, anti-nephrin autoantibodies have been found in up to one-third of patients with minimal change disease [7], suggesting that podocytes themselves are the targets of immune complexes.

Our results not only showed that knockout of FcRn in podocytes affects lysosomal function but also that signaling pathways differ in WT and FcRn KO podocytes after treatment with IgG or immune complexes. We found differential upregulation of phospho-Akt (S473) or phospho-ERK1/2 in WT versus KO podocytes depending on the treatment conditions (ICs versus IgG or NP-ovalbumin), suggesting that podocytes have the ability to distinguish between different types of exogenous proteins and between IgG alone versus immune complexes.

Both the Akt and ERK1/2 pathways have been shown to be involved in signaling and lysosomal activation or antigen presentation after an immune challenge [24, 26, 36]. Our findings that KO podocytes showed little or no increase in phospho-Akt or phospho-ERK1/2 expression compared to baseline after treatment with ICs whereas WT podocytes had increased expression of these signaling mediators after IC treatment support the hypothesis that FcRn-mediated trafficking of immune complexes in podocytes modulates the podocyte response to immune stimuli and that lack of FcRn dampens the response to an immune insult resulting in less metabolically active and less immunogenic lysosomes. The role of podocyte lysosomal function in glomerular health and disease is still an active matter of investigation. How lysosomal dysfunction causes podocyte loss and proteinuria in glomerulonephritis remains unknown but several studies have found dysregulation of key lysosomal components in glomerular disease. Yamamoto-Nonaka et al. found that cathepsin D expression (a lysosomal protease) was upregulated in podocytes in patients with minimal change disease (which is mediated at least in part by immune mechanisms) but not in those with focal segmental glomerulosclerosis [37]. Cathepsin D was also found to be upregulated in membranous nephropathy, another immune-mediated kidney disease [38]. In antineutrophil cytoplasmic antigen (ANCA)- mediated vasculitis, an aggressive autoimmune disease which frequently causes crescentic glomerulonephritis, a subset of patients were found to have autoantibodies to LAMP2 [39], suggesting that lysosomes may play a role in the pathogenesis of this disease. Studies in mice have also confirmed that lysosomal dysfunction plays a pathogenic role in immune-mediated kidney diseases [40].

Taken together, our work demonstrates that FcRn- mediated trafficking of immune complexes modulates podocyte function in vitro and that cultured podocytes respond differently

to IgG versus immune complexes. A mechanistic understanding of how podocytes handle immune complexes may allow for the creation of targeted therapies to treat immune-mediated glomerulonephritis.

## Supporting information

**S1 Fig. Uncropped western blot gels for western blot images in Figs 4 and 6.** For Fig 4, uncropped gels are shown for LAMP1 at 4 hours, cathepsin B at 4 hours, LAMP1 at 48 hours, cathepsin B at 48 hours and corresponding beta-actin for each gel. For Fig 6, uncropped gels are shown for NP-ovalbumin (NP-Ova), IgG and immune complex (IC)- treated WT and KO podocytes for phospho-Akt (pAkt), total Akt, phosphor-ERK1/2, total ERK1/2 and corresponding beta-actin levels.
(PDF)

## Author Contributions

**Conceptualization:** Judith Blaine.

**Formal analysis:** George Haddad, Judith Blaine.

**Methodology:** George Haddad, James Dylewski, River Evans, Linda Lewis, Judith Blaine.

**Supervision:** Judith Blaine.

**Writing – original draft:** Judith Blaine.

**Writing – review & editing:** George Haddad, James Dylewski, River Evans, Judith Blaine.

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
