## [Decision Letter · Decision Letter 0]

22 Feb 2023

PONE-D-23-02340Knockout of the neonatal Fc receptor alters immune complex trafficking and lysosomal function in cultured podocytesPLOS ONE

Dear Dr. Blaine,

Thank you for submitting your manuscript to PLOS ONE. After careful consideration, we feel that it has merit but does not fully meet PLOS ONE’s publication criteria as it currently stands. Therefore, we invite you to submit a revised version of the manuscript that addresses the points raised during the review process.

Your manuscript is well written and the experiments are scientifically sound, however, in order to make it even better, the discussion needs to be improved as stated by the reviewer.

We look forward to receiving your revised manuscript.

Kind regards,

Boyan Grigorov

Academic Editor

PLOS ONE

“This work was funded by NIH National Institute of Diabetes and Digestive and Kidney Diseases R01 DK104264 to JB.”

“This work was funded by NIH R01 DK104264 to JB.”

“This work was funded by NIH National Institute of Diabetes and Digestive and Kidney Diseases R01 DK104264 to JB.”

Reviewers' comments:

Reviewer's Responses to Questions

**Comments to the Author**

1. Is the manuscript technically sound, and do the data support the conclusions?

Reviewer #1: Yes

2. Has the statistical analysis been performed appropriately and rigorously? 

Reviewer #1: Yes

3. Have the authors made all data underlying the findings in their manuscript fully available?

Reviewer #1: Yes

4. Is the manuscript presented in an intelligible fashion and written in standard English?

Reviewer #1: Yes

5. Review Comments to the Author

Reviewer #1: Renal function and pathological mechanisms of a number of immune-mediated kidney injuries are current medical problems of immediate importance. Indeed, in vitro experiments illuminate many of these mechanisms and suggest real pathological mechanisms involving the immune system as cells and sub-cellular components. The experimental staging is very well and precisely executed. I have no objections regarding the results and their analysis, many microscopic setups were done with different stains and cell signaling and proliferation analyzes were well done. What I miss in the discussion is the lack of sufficient speculation about the relationship of the experiments done and the results obtained to real events in autoimmune processes and their direct impact on proteinuria. Of course, only one in vivo experiment would give a real answer to this question, but the obtained experimental data are enough to form a complete pathological picture and process occurring in immune-complex mediated diseases. Perhaps a small discussion to put together all the results obtained would give a more complete picture and the actual merits of the paper.

6. PLOS authors have the option to publish the peer review history of their article (what does this mean?). If published, this will include your full peer review and any attached files.

Reviewer #1: No

---

## [Author Response · Author response to Decision Letter 0]

31 Mar 2023

We have made the changes requested by the Editorial office and also added to the Discussion as requested by the reviewer to provide greater biologic and clinical context for our results. Please see Response to Reviewers and redlined and final manuscript for details.

---

## [Editor Report · Decision Letter 1]

5 Apr 2023

Knockout of the neonatal Fc receptor alters immune complex trafficking and lysosomal function in cultured podocytes

PONE-D-23-02340R1

Dear Dr. Blaine,

We’re pleased to inform you that your manuscript has been judged scientifically suitable for publication and will be formally accepted for publication once it meets all outstanding technical requirements.

Kind regards,

Boyan Grigorov

Academic Editor

PLOS ONE
---

## [Editor Report · Acceptance letter]

10 Apr 2023

PONE-D-23-02340R1 

Knockout of the neonatal Fc receptor alters immune complex trafficking and lysosomal function in cultured podocytes 

Dear Dr. Blaine:

I'm pleased to inform you that your manuscript has been deemed suitable for publication in PLOS ONE. Congratulations! Your manuscript is now with our production department. 

Kind regards, 

on behalf of

Dr. Boyan Grigorov 

Academic Editor

PLOS ONE